# Evaluation of nucleosome concentrations in healthy dogs and dogs with cancer

**Heather Wilson-Robles**[1]*, **Tasha Miller**[1], **Jill Jarvis**[1], **Jason Terrell**[2], **Nathan Dewsbury**[2], **Terry Kelly**[2], **Marielle Herzog**[3], **Thomas Bygott**[3], **Nathalie Hardat**[3], **Gaetan Michel**[3]

1 Small Animal Clinical Sciences Department, College of Veterinary Medicine, Texas A&M University, College Station, Texas, United States of America, 2 Volition America & Volition Veterinary Diagnostic Development, Austin, Texas, United States of America, 3 Belgian Volition SPRL, Isnes, Belgium

* hwilson@cvm.tamu.edu

**Data Availability Statement:** All relevant data are within the paper.

**Funding:** Funding for materials and author salaries was provided by Belgian Volition SPRL. HWR and TM received salary from these funds. JT, TK, ND, MH, TB, NH and GM are employees of Belgian

## Abstract

### Introduction

Nucleosomes consist of small fragments of DNA wrapped around a histone octamer core. Diseases such as cancer or inflammation lead to cell death, which causes fragmentation and release of nucleosomes into the blood. The Nu.Q™ technology measures circulating nucleosome levels and exploits the different compositions of cancer derived nucleosomes in blood to detect and identify cancer even at early stages. The objectives of this study are to identify the optimal sample type for the Nu.Q™ H3.1 assay and to determine if it can accurately detect nucleosomes in the blood of healthy canines as well as those with cancer.

### Materials and methods

Blood samples from healthy canine volunteers as well as dogs newly diagnosed with lymphoma were used. The blood was processed at a variety of times under a variety of conditions to determine the most reliable sample type and conditions, and to develop an appropriate processing strategy to ensure reliably accurate results.

### Results

Nucleosomes could be detected using a variety of sample collection and processing protocols. Nucleosome signals were highest in EDTA plasma and serum samples and most consistent in plasma. Samples should be processed within an hour of collection. Experiments showed that samples were able to withstand several freeze thaw cycles. Processing time and tcollection tube type did affect nucleosome detection levels. Finally, significantly elevated concentrations of nucleosomes were seen in a small cohort of dogs that had been newly diagnosed with lymphoma.

### Conclusions

When samples are collected and processed appropriately, the Nu.Q™ platform can reliably detect nucleosomes in the plasma of dogs. Further testing is underway to validate and optimize the Nu.Q™ platform for veterinary use.

Volition & Volition America. The URL to the Belgian Volition website is: https://volition.com/. Some additional funding for HWR and TM salaries was provided by the Fred and Vola Palmer Chair of Comparative Oncology held by HWR.

**Competing interests:** I have read the Journal's policy and the authors of this manuscript have the following competing interests: JT, ND, TK, MH, TB NH and GM are employees of Belgian Volition & Volition America, which have patents covering Nu. Q technology and are developers of Nu.Q™ assays. Volition Veterinary is a joint venture between Belgian Volition and Texas A&M University. HWR is a paid consultant of Volition Veterinary. TM and JJ have no conflicts of interest to declare. Additional salary support for TM was provided by the Fred and Vola Palmer Chair in Comparative Oncology held by HWR. The Palmers did not play a role in the study design, data collection and analysis, decision to publish, or preparation of the manuscript and only provided financial support for the authors' salaries (T.M.). This does not alter our adherence to PLOS ONE policies on sharing data and materials.

## Introduction

Nucleosomes are small fragments of chromosomes [1] that are composed of a 147 bp segment of DNA wrapped around 4 core histones present in duplicate for a total of 8 histones. These core histones are highly conserved between eukaryotic species and are relatively invariant between lower species, such as yeast, and mammals, including humans [2, 3].

Nucleosomes have many functions in the cell. They provide the framework for chromatin assembly that is required for chromatin compaction, protect DNA from damaging agents and are critical for the stable repression of certain genes by restricting binding of transcription factors to DNA sequences. Nucleosomes alter their structure allowing for access to DNA during transcription, repair and DNA synthesis. Furthermore, nucleosomes act as a framework where a variety of epigenetic signals are laid [4]. While nucleosomes are present in all mammalian cells, they can also be detected circulating in blood, where they are most commonly released by activated or dying white blood cells [5, 6]. Large numbers of nucleosomes are released into the blood of humans and animals suffering from severe inflammation or trauma [7–9]. These small cell free (cf) DNA molecules have been shown to have immunostimulatory roles that differ from that of free circulating histones or double stranded cell free DNA(ds-cfDNA) [10]. The immunostimulatory effects of nucleosomes appear to be cell type dependent and may rely on specific surface markers such as DAMP high-mobility group box 1 (HMGB1) or the receptor for advanced glycation end products (RAGE), and require apoptosis rather than necrosis for activation [10].

Elevated concentrations of nucleosomes have been identified in the blood of cancer patients. A study by Rasmussen et al [11] demonstrated that elevated nucleosome levels could be detected reliably. Nucleosomes have also been found to improve the detection of pancreatic cancer using serum when compared to the common blood marker, carbohydrate antigen 19–9 (CA 19–9) in a study published in 2015 [12]. Though there are no published studies specifically describing cancer detection using nucleosomes in dogs, several publications have described the utility of cfDNA [13–17].

The current manuscript aims to define an optimized technique for isolating and analyzing this important cfDNA component and better understand circulating nucleosomes in healthy canines and using the Nu.Q™ H3.1 ELISA assay. This assay is the first of many developed to analyze nucleosomes in both humans and dogs. We further show that similar to humans, elevated nucleosome levels are present in canines with cancer compared to healthy controls.

## Materials and methods

All animal studies were approved by the Texas A&M University Institutional Animal Care and Use Committee (AUP #2019–0211 CA and AUP #-2017-0350).

Seven healthy dogs were recruited for up to 3 separate blood draws. In order to be eligible dogs needed to be healthy, over 3 years of age, weigh more than 10 kg and not be pregnant. Dogs over the age of 3 were chosen as they best represent the target group of clinical cancer patients for which this assay has been developed. The dogs were a variety of breeds (pure bred dogs included 1 Australian cattle dog, 1 Australian shepherd, the rest were mixed breed dogs) with 5 spayed females and 2 neutered males. The dogs ranged in age from 4 years to 14 years of age and all dogs had good body condition scores of 4–6 on a 9-point scale. Not all dogs participated in every assay, but a minimum of 5 dogs were used in all assays.

The capture antibody for the Nu.Q™ H3.1 assay (Active Motif, Carlsbad, CA) was validated for use in canines using Mass Spectrometery by Spectrus Corp (Beverly, MA). Briefly, two plasma samples obtained from canines newly diagnosed with lymphoma were used. Baseline nucleosome concentrations were determined using the Nu.Q™ H3.1 ELISA assay following the

manufacturer's directions (see below). Immunoprecipitation was performed on the samples using beads coated with the anti-H3.1 capture antibody. Samples were incubated with the beads at room temperature for 1 hour in a rotating mixer and separated with a magnet. Samples were washed twice with PBS and the assay buffer. The immunoprecipitated proteins were resuspended in the assay buffer and treated with 2 μg of trypsin overnight at 37˚C and boosted with another 2 μg of trypsin in the morning. The beads were removed with a magnet and the supernatant was acidified with TFA to a final concentration of 1% (v/v) and placed in HPLC vials for analysis.

All samples were tested using the Nu.Q™ H3.1 assay. This is an enzyme-linked immunosorbent assay (ELISA) with a capture antibody directed at histone 3.1 and nucleosome specific detection antibody [18]. Assays were performed according to the manufacturer's instructions. Briefly, a standard curve was generated using the positive control stock (recombinant H3.1 nucleosomes) provided. The nucleosomes were bound to the detection antibody and the plates were washed 3 times using the provided 1x wash buffer. Twenty microliters of each undiluted sample were pipetted in duplicate into wells on the 96 well plates. Next, 90uL of the assay buffer was added to each well. The plate was covered with sealing film and incubated on an orbital shaker for 2.5 hours at 700 rpm. Plates were then emptied and washed 3 times using the 1x washing buffer. Next, 100 uL of the detection antibody was added to each well, the plate was resealed and incubated for 1.5 hours on the orbital shaker. The plates were then washed as described above. Streptavidin HRP conjugate was incubated for 30 min in each well and washed before applying the colorimetric substrate solution and incubating the plates in the dark for 20 min. A stop solution was added to the wells and the plates were read on a plate reader at 405 nm (BioTek Synergy H1 plate reader, BioTek Instruments, Winooski, VT). The standard curve was linearized and fitted to a 5-parameter logistic curve using statistical software (Graphpad Software, version 8, San Diego, CA).

In order to determine how processing times affected nucleosome concentrations in canine blood samples, the first blood collection included 20 mL of blood from 6 dogs separated into EDTA plasma (lavender top) or serum tubes (red top) (Becton, Dickinson and Company, Franklin Lakes, NJ). Nine time points were evaluated from each sample type: time 0, 15 min, 30 min, 45 min, 1 hour, 2 hours, 4 hours, 8 hours and 24 hours. Samples were left at room temperature until their designated processing time. When processed, samples were centrifuged at room temperature at 3000xg for 10 min. Serum or plasma was then immediately removed, placed in pre-labeled cryovials and frozen at -80˚C to run in batches. All samples were run in duplicate.

To evaluate which type of plasma or serum sample gave the most reliable results, a second batch of 20 mL of blood was collected from the same 6 healthy volunteer dogs 2 months after the first blood collection. This blood was separated into a simple serum tube (red top), a serum separator tube (yellow top), EDTA plasma (lavender top) and sodium citrate plasma (blue top) (Becton, Dickinson and Company, Franklin Lakes, NJ). Samples were processed at time 0, 30 minutes and 60 minutes after the blood draw. These times were chosen based on the results of the first assay. Samples remained in their designated tubes at room temperature until their specified processing time. When processed, samples were centrifuged at room temperature at 3000xg for 10 min. Serum or plasma was then immediately removed, placed in pre-labeled cryovials and frozen at -80˚C to run in batches. All samples were run in duplicate.

In order to determine if temporary storage conditions associated with different shipping methods can affect the concentration of nucleosomes, identically processed samples from 5 dogs (EDTA and citrate plasma) were packaged in a box either on ice or at room temperature and left on the counter overnight. Samples were processed 24 hours later using the Nu.Q™ H3.1 ELISA assay. Samples were run in duplicate and compared for possible differences.

In order to determine how multiple freeze thaw cycles affect nucleosome concentrations, an additional 15 mL of blood was collected from 7 healthy volunteers 2 months after the second sample collection and divided into three aliquots (one dog in the previous assay was replaced by a new dog and all dogs were available for this blood draw). The samples were centrifuged immediately at 3000xg for 10 min at room temperature and the plasma was divided into cryovials. Control (time 0) samples were analyzed immediately and the remaining sample was stored at -80°C for future analysis. Frozen aliquots were thawed and refrozen up to 5 times analyzing the nucleosome concentrations in each sample at each freeze thaw cycle. All samples were run in duplicate.

An additional 3 mL of blood was taken from 6 healthy dog volunteers on two separate occasions. The first blood collection was performed while animals were fasted and the second after a meal. The samples were immediately centrifuged at room temperature at 3000xg for 10 min and the plasma was collected and stored at -80°C. Duplicate samples were analyzed in batches.

To determine the effects of processing times on cancer derived nucleosomes, 3 mL of blood was drawn from 13 client owned canines with lymphoma (AUP #-2017-0350). All patients were newly diagnosed and naïve to treatment. Following collection, samples were aliquoted into 5 tubes and processed immediately, at 30 minutes, 1 hour, 2 hours and 24 hours after collection. Samples were kept at room temperature until the designated processing time. Samples were compared to the healthy dogs from Fig 2. After processing the plasma was collected and stored at -80°C until analyzed.

The optical density (OD) values determined by the ELISA for each sample were plotted against a standard curve of known nucleosome concentrations. All concentrations were interpolated using an asymmetric sigmoidal curve with a five-parameter logistic equation (5PL) where X = Concentration.

When evaluating the processing time points and the sample type, a correlation matrix was calculated containing the correlations between the results at each possible pair of time points. This was done using Pearson's correlation coefficient using concentration values and Kendall's Tau coefficient, based on concordance between pairs. Both measures take values between -1 and 1. The results presented are the correlations between each time point and time zero. Both methods led to the same conclusion regarding the maximum time before centrifugation. To assess the question of whether there is a systematic bias over time, scatterplots were produced for each time point versus time zero and the differences tabulated. This part of the analysis was conducted using the statistical programming language R (R Core Team (2017). R: A language and environment for statistical computing. R Foundation for Statistical Computing, Vienna, Austria. URL https://www.R-project.org). Graphs were produced using ggplot2 (H. Wickham. ggplot2: Elegant Graphics for Data Analysis. Springer-Verlag New York, 2016).

For data sets containing only two conditions, such as the evaluation of storage of samples at room temperature or on ice or fasting versus fed conditions a Wilcoxon signed rank test was used to compare the medians of the data sets. For data sets where multiple conditions were compared, such as the multiple freeze thaw cycles and the lymphoma versus healthy cases, a two-way ANOVA for repeat measures with a Tukey's multiple comparisons test was performed. This part of the analysis was performed using GraphPad Prism version 8.0.0 for Macintosh, GraphPad Software, San Diego, California USA, www.graphpad.com.

## Results

### The Nu.Q™ H3.1 assay is specific for canine nucleosomes

A total of 339 proteins were identified during the mass spectrometry analysis between two samples, including peptides for H3.1 and other histone proteins. Baseline nucleosome

concentrations in plasma were 960 ng/mL and 480 ng/mL and 70 ng/mL and 40 ng/mL after immunoprecipitation resulting in 93% and 91% nucleosome depletion in each sample, respectively.

## Optimal sample processing times vary depending on the sample type used

Consistent with what has been observed for human samples (data not shown) we found that serum samples were far more variable than plasma. There were large variations in nucleosome concentrations even within 15 min in at least half of the dogs' serum samples (Table 1, Fig 1). The time point with the least amount of variation when compared to time 0 was 120 minutes after collection with a Pearson's correlation coefficient of 0.90. The second highest correlation timepoint was at a processing time of 30 min. The largest difference in serum nucleosome levels was seen at 24 hours with mean and median percent differences of 50% and 25.8%, respectively. The majority of plasma samples had stable nucleosome levels as long as they were processed within 60 min of collection (Table 2, Fig 2). The highest mean nucleosome

**Table 1. Correlation between each time point and time zero, demonstrating the variation in serum nucleosome concentrations.**

| Processing time | Pearson's Correlation | Kendall's Tau |
|---|---|---|
| Time 0 | 1.00 | 1.00 |
| 15 min | 0.29 | 0.20 |
| 30 min | 0.88 | 0.60 |
| 45 min | 0.42 | 0.40 |
| 60 min | 0.80 | 0.40 |
| 120 min | 0.90 | 0.80 |
| 4 hours | 0.68 | 0.40 |
| 8 hours | 0.14 | 0.40 |
| 24 hours | 0.70 | 0.40 |

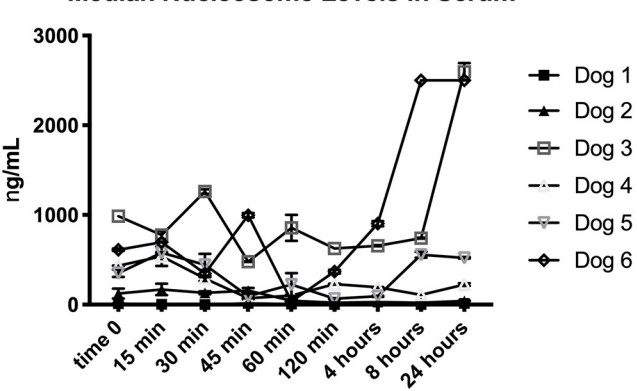

(A)

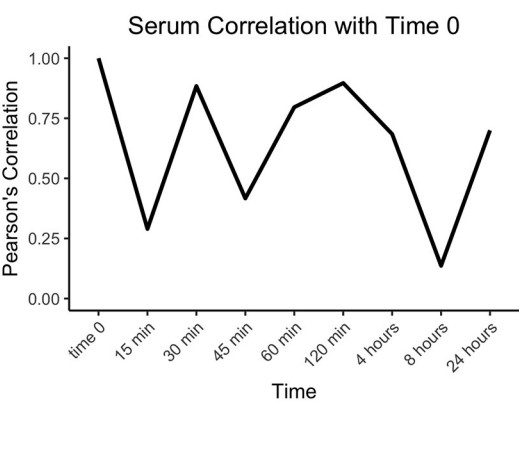

(B)

**Fig 1. Nucleosome concentrations in canine serum.** A. Median nucleosome concentrations (ng/mL) in serum for all dogs. There is a noticeable amount of variation after 15 minutes in nearly all dogs with the exception of dogs 1 and 2 whose nucleosome concentrations did not change appreciably at any time point. B. Graphical representation of the Pearson's correlation coefficients for this data set. There was very little correlation over time between the serum samples.

**Table 2. Correlation between each time point and time zero, demonstrating the variation in plasma nucleosome concentrations.**

| Processing time | Pearson's Correlation | Kendall's Tau |
|---|---|---|
| Time 0 | 1.00 | 1.00 |
| 15 min | 0.98 | 0.80 |
| 30 min | 0.98 | 0.80 |
| 45 min | 0.96 | 0.80 |
| 60 min | 0.96 | 0.80 |
| 120 min | 0.79 | 0.60 |
| 4 hours | 0.77 | 0.60 |
| 8 hours | 0.46 | 0.20 |
| 24 hours | 0.65 | 0.20 |

Both measures of correlation remain high until 60 minutes and then reduce for longer processing times.

concentrations were recorded for most dogs at times 0, 8 hours and 24 hours. The largest percent changes seen in nucleosome concentrations when compared to time 0 were between 4–24 hours with mean percent changes ranging from -20.1–45.6% and median percent changes ranging from -43.3–65.8%. The Pearson's correlation coefficients showed much higher consistency than serum, being at 0.96 or above for the 15 min, 30 min and 60 min time points (Table 2, Fig 2B). To check for systematic bias, a series of scatterplots were produced comparing each time point for both serum and plasma readings to the time zero readings. The data in Table 3 and Fig 3 show no consistent systematic bias in plasma. Similar results were seen in serum (data not shown).

## Plasma provides more stable nucleosome concentrations than serum

A total of 4 sample types were tested with a variety of processing times up to 1 hour after collection. Extended processing times were not evaluated due to the wide variability seen in the previous experiment. Plasma provided the most consistent nucleosome concentrations

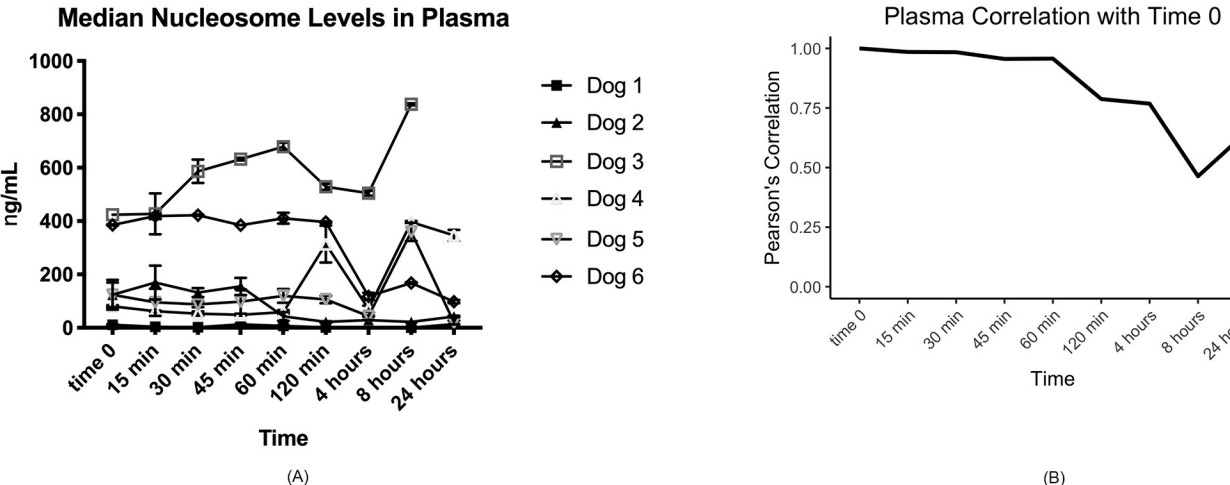

(A)

(B)

**Fig 2. Median nucleosome concentrations in canine plasma.** A. Median plasma concentrations (ng/mL) in all dogs over time. With the exception of Dog 3, most dogs have very stable nucleosome concentrations for the first 60 minutes before processing. B. Graphical representation of the Pearson's correlation coefficients of all plasma data point for the 6 dogs. There is much better correlation of the samples to time 0 control in this data set.

**Table 3. Changes in plasma measurements compared to time zero.**

| | ng/mL | Percentage difference from time 0 | | | | |
|---|---|---|---|---|---|---|
| | Time 0 | 15 min | 30 min | 45 min | 60 min | 2 hours |
| Dog1 | 11.3 | -69% | -84% | 18% | -39% | -88% |
| Dog2 | 123.5 | 37% | 7% | 26% | -66% | -82% |
| Dog3 | 424.0 | 1% | 38% | 49% | 60% | 25% |
| Dog4 | 80.2 | -23% | -34% | -39% | -27% | 292% |
| Dog5 | 123.7 | -23% | -29% | -21% | -3% | -14% |
| Dog6 | 385.1 | 9% | 10% | 0% | 7% | 3% |
| Average | 191.3 | -11% | -15% | 5% | -11% | 22% |

between samples and there was no difference in the consistency of the sample type over time between the citrate and EDTA plasma samples. The serum red top tubes, which contain no additives, were the most variable of the serum samples (Fig 4), though there was no statistically significant difference between the time points for any of the serum samples.

Short-term storage at room temperature or on ice does not significantly affect nucleosome concentrations. Plasma samples (EDTA and sodium citrate) were evaluated using the Nu.Q™ H3.1 assay after being packaged for shipping either at room temperature or on ice overnight. The median concentration of the EDTA samples stored at room temperature was 112.8 ng/mL and for those stored on ice was 76.35 ng/mL. The two were not statistically different ($p = 0.0625$). The mean nucleosome concentration in the citrate plasma samples stored at room temperature was 74.1 ng/mL and for those on stored on ice was 23.53 µg/mL (Fig 5). These two sets of samples were also not statistically different ($p = 0.125$) either, however, in all sample types, those stored on ice had values that were more consistent with the time 0 concentrations for these sample times seen in Fig 4.

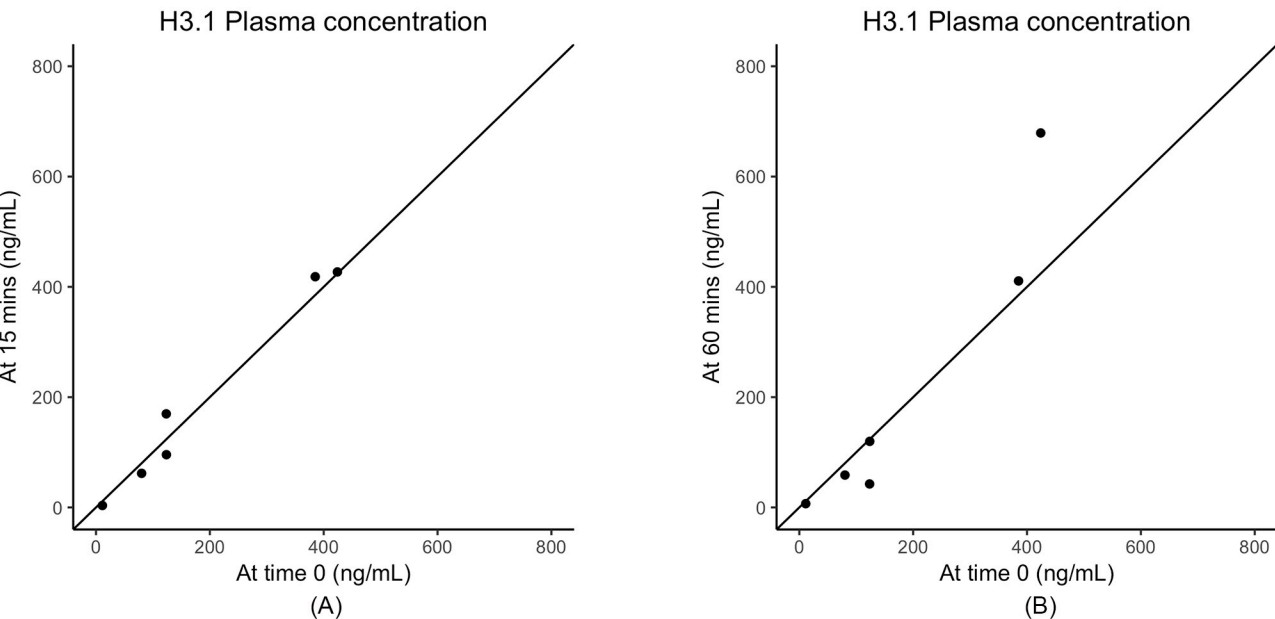

**Fig 3. Scatterplots of nucleosome concentrations in canine plasma.** A. After processing time of 15 minutes compared to time zero. B 60 minutes compared to time zero.

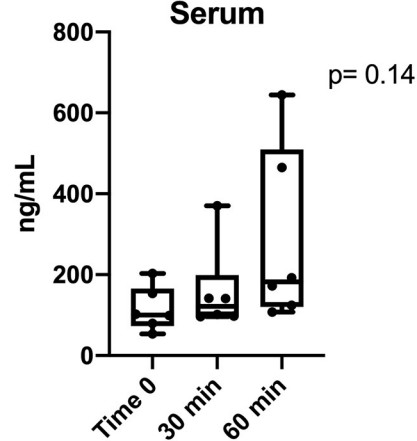

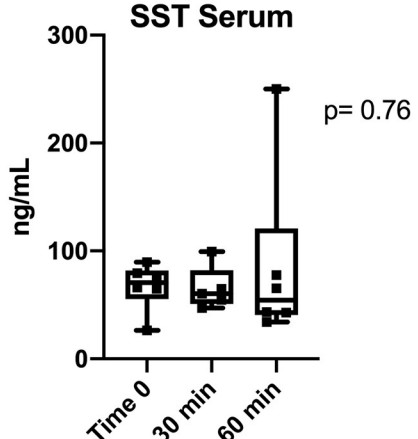

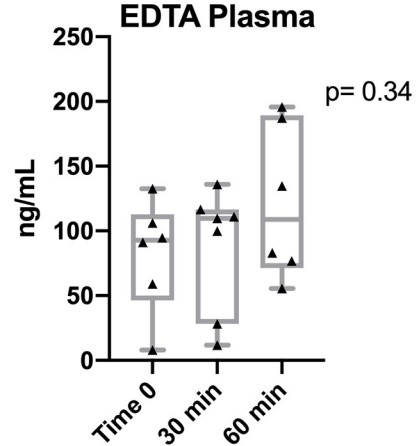

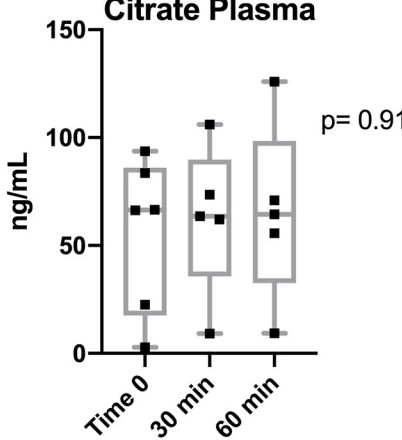

**Fig 4. Consistency of nucleosome levels between various sample types.** Median nucleosome concentrations in serum (A & B) and plasma (C & D) samples from 6 healthy canine volunteers. Plasma samples had more consistent nucleosome concentrations than serum samples. Red top tube and EDTA tube samples contained higher nucleosome concentrations than SST and citrate tubes when comparing samples from the same dogs.

### Nucleosome concentrations are not significantly affected by multiple freeze/thaw cycles

Duplicate samples from 7 healthy canine volunteers were evaluated after 5 freeze-thaw (FT) cycles to determine if repeated freeze-thaw cycles would affect the nucleosome concentrations in the plasma. The mean nucleosome concentrations for all dogs are reported in Table 4. There were no significant differences noted between any of the cycles, though mean concentrations were routinely higher in FT cycle 1 for all dogs. Four of the 6 dogs had very stable nucleosome concentrations (< 50 ng/mL change) during all of the freeze thaw cycles. However, samples from dogs 3 and 4 had a noticeable decrease in nucleosome concentration at the 3rd or 4th FT cycle (Fig 6).

### Fasting significantly affect mean nucleosome concentrations

Six canine volunteers were either fasted for 10–12 hours or fed within 2 hours before blood collections. Samples were analyzed and the medians for all dogs were compared. The median

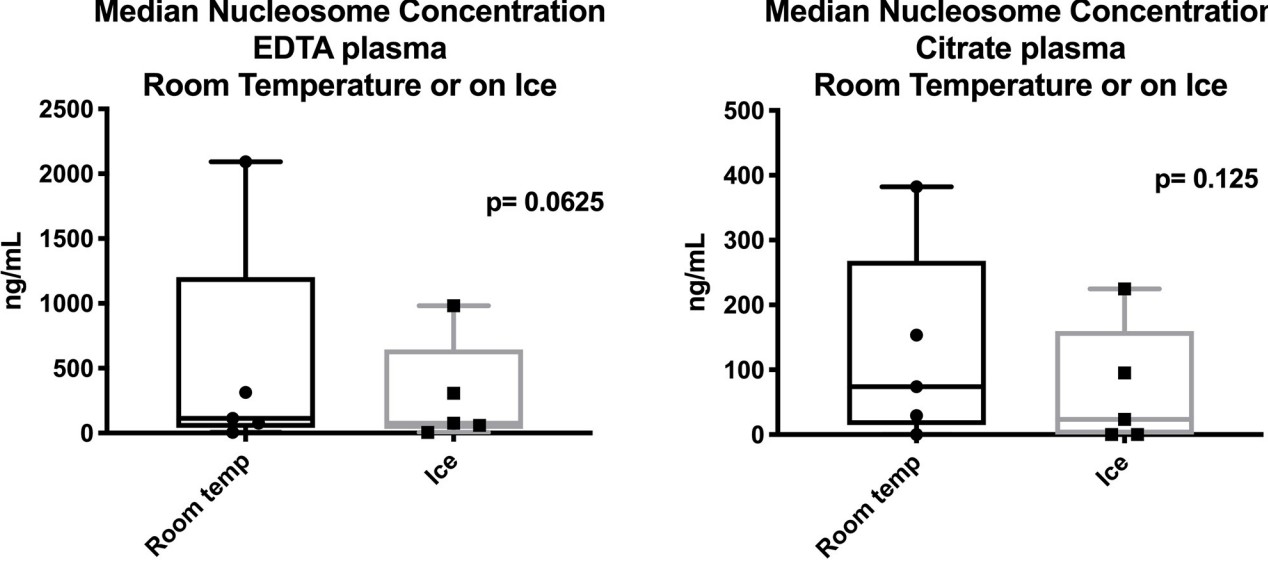

**Fig 5. Effects of short-term storage on nucleosome concentrations.** Though median nucleosome concentrations (ng/mL) were consistently higher when kept at room temperature overnight, there was no significant difference between the two conditions.

concentration of nucleosomes for all dogs fasted was 65.5 ng/mL (range 3.0–788.0 ng/mL) and for all dogs after feeding was 62.75 ng/mL (range 1.0–1191.0) (p = 0.0312). When individual dogs were compared, there was a noticeable difference between fasting and fed samples for dogs 2 and 3. Both of these dogs had noticeably higher nucleosome concentrations in plasma after eating (Fig 6). Given the small numbers of dogs in this group and the wide variability in nucleosome concentrations seen, a larger study with additional dogs is be needed in order to understand the impact of feeding on circulating nucleosome levels.

Nucleosome concentrations are stable at a variety of processing times in dogs with lymphoma.

Thirteen dogs with newly diagnosed lymphoma (12 multicentric lymphoblastic lymphomas and 1 indolent T cell lymphoma) were recruited for this cohort. EDTA plasma samples were processed over a variety of time points and analyzed for median nucleosome concentrations. There was no significant difference between the mean or median concentrations for this group at any of the processing time points. The mean nucleosome concentrations at time 0, 30 min, 60 min, 120 min and 24 hours were 661.2 ng/mL, 640.9 ng/mL, 638.8 ng/mL, 702.3 ng/mL and 537.1 ng/mL respectively (Fig 8, Table 5). Nucleosome concentrations in lymphoma samples (median 590 ng/mL for all dogs at all timepoints) were significantly higher at all time points than age matched healthy control dogs (median 116.5 ng/mL for all dogs at all time points) with a p value of 0.0079 (Fig 7).

**Table 4. Mean concentrations (ng/mL) of nucleosomes in EDTA plasma after 5 freeze thaw cycles.**

|  | FT 1 | FT 2 | FT 3 | FT 4 | FT 5 |
|---|---|---|---|---|---|
| **Mean** | 67.19 | 60.38 | 51.98 | 56.57 | 58.47 |
| **SD** | 37.68 | 55.33 | 54.81 | 38.78 | 41.69 |
| **SEM** | 14.24 | 20.91 | 20.72 | 15.83 | 17.02 |
| **P value** |  | 0.8898 | 0.5315 | 0.1624 | 0.2860 |

P values were calculated comparing additional freeze thaw cycles to the first freeze thaw cycle.

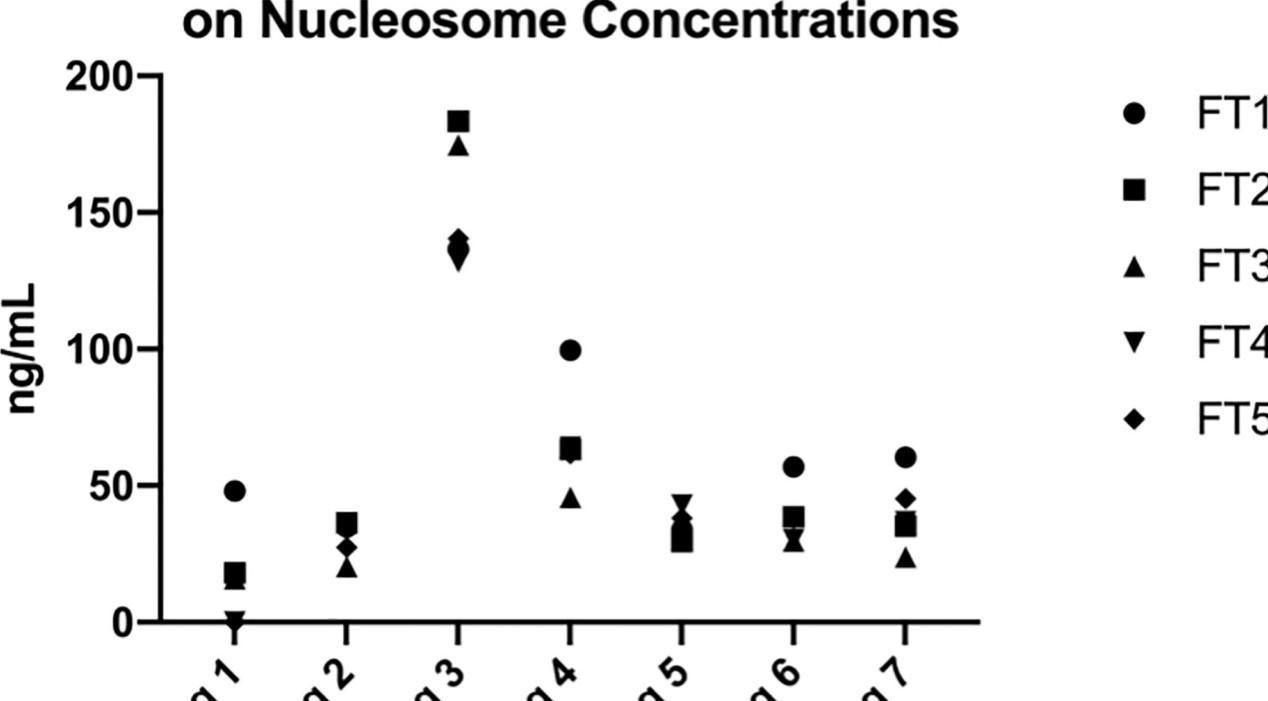

**Fig 6. Effects of freeze/thaw cycles on nucleosome concentrations.** Mean nucleosome concentrations (ng/mL) at each freeze thaw cycle for all dogs. There was very little change (<50 ng/mL) in mean nucleosome levels for 5 of 7 dogs. However, dogs 3 and 4 did display more variability between samples.

## Discussion

Nucleosomes contain DNA wrapped around an octamer containing histone sub-units, H3, H4, H2A and H2B. H3 has two main variants, H3.1 and H3.3 [19]. We targeted the H3.1 sub-unit and expected the H3.1 antibody would bind the canine histone protein due to the high degree of homology between the two species (>96%). We were also able to identify all four histone units in the immunoprecipitated protein samples from the two dogs with lymphoma suggesting that we were able to isolate entire nucleosomes rather than individual histones in the plasma. The depletion of nucleosomes in the two samples after immunoprecipitation demonstrates the high affinity of the antibody for the canine histone. Elevated concentrations of nucleosomes have previously been identified in dogs with a variety of diseases including sepsis, trauma, septic peritonitis and immune mediated hemolytic anemia; however, to the authors'

**Table 5. Mean nucleosome concentrations (ng/mL) for samples processed at a variety of times with SD and SEM for 13 dogs newly diagnosed with lymphoma.**

|  | Time 0 | 30 min | 60 min | 120 min | 24 hours |
|---|---|---|---|---|---|
| **Mean** | 661.2 | 640.9 | 638.8 | 702.3 | 537.1 |
| **Std. Deviation** | 841.9 | 855.7 | 863.7 | 882.9 | 683.1 |
| **Std. Error of Mean** | 233.5 | 237.3 | 239.5 | 244.9 | 189.4 |
| **Percent Change** |  | 3.1% | 3.4% | 5.9% | 19.8% |

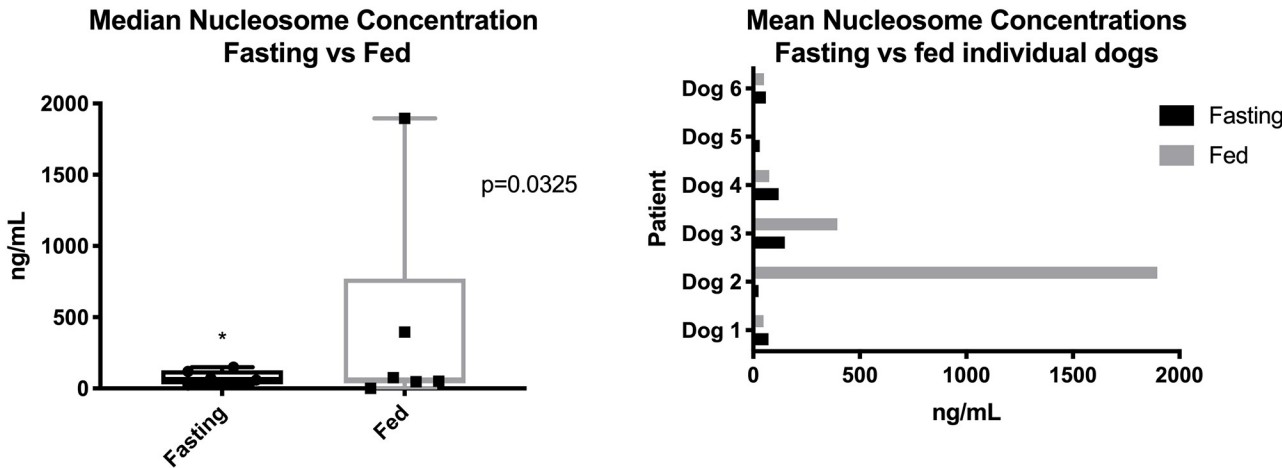

**Fig 7. Effects of fasting on nucleosome concentrations.** A. Median nucleosome levels (ng/mL) for all dogs after a meal or after ≥ 10 hours fasting. The fasting samples were significantly lower than the fed samples. B. Mean nucleosome levels (ng/mL) in each individual dog after a meal or after ≥ 10 hours of fasting.

knowledge this is the first time the nucleosome concentrations have been defined in healthy dogs or dogs with cancer [7–9, 20, 21]. In general, the concentration of circulating nucleosomes in healthy dogs is low with medians ranging from 40–100 ng/mL. This is significantly lower than the concentration seen in the dogs with lymphoma reported in this study with

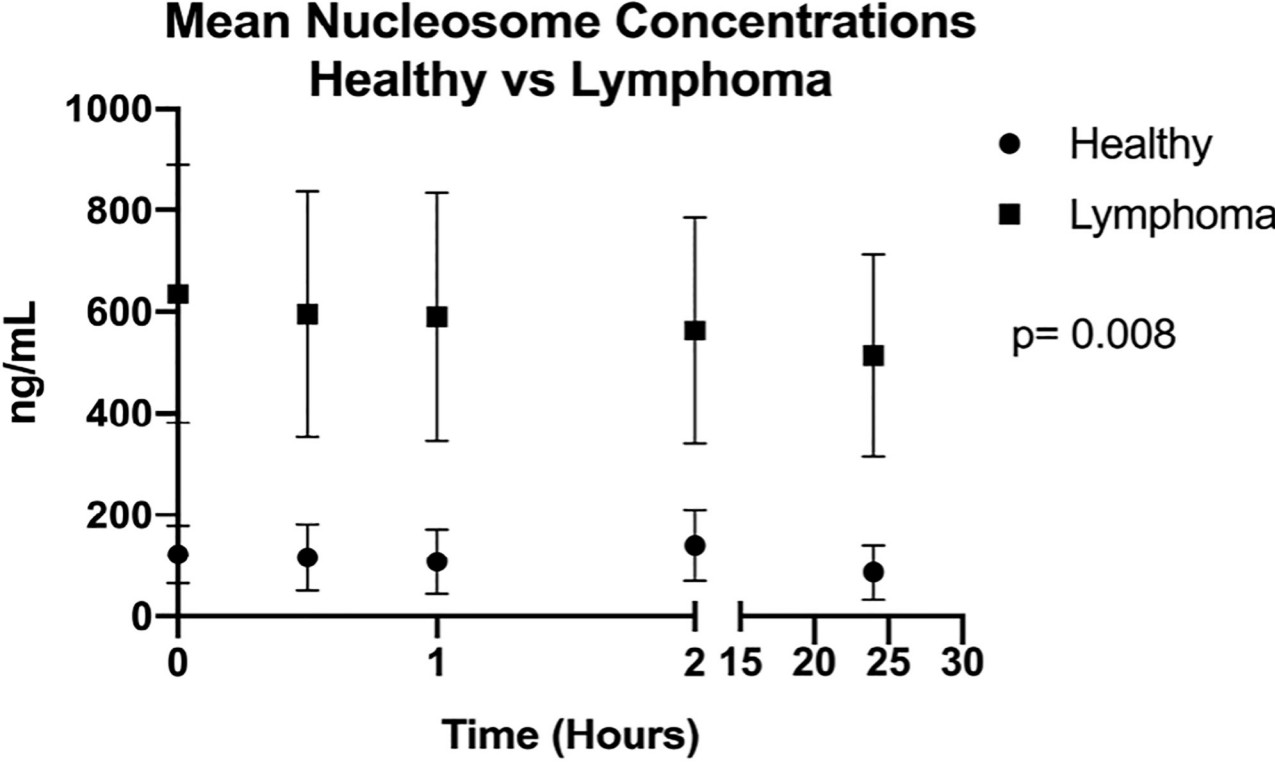

**Fig 8. Stability of cancer-associated nucleosomes in dogs with lymphoma.** Mean nucleosome concentrations from 6 healthy dogs (as pictured in Fig 2) and 13 dogs newly diagnosed with lymphoma. The 6 healthy dogs were not fasted whereas the dogs with lymphoma were; however, there is still a significant elevation in median nucleosome concentrations at all time points for the dogs with lymphoma.

median of 590 ng/mL. Comparisons between the concentrations of nucleosomes reported here and those in other reports of inflammation and control groups are difficult due to the fact that nucleosome concentrations have previously been reported only in arbitrary units [7–9, 20–22].

When evaluating nucleosome concentrations in plasma and serum at a variety of processing times plasma is more stable than serum. This finding is in agreement with findings reported by Goggs, 2019. Based on the results reported here, optimal processing times for plasma are within 60 minutes of collection, though processing within 30 minutes of collection are ideal. The optimal processing time for serum samples was set at 15 minutes, however, as these samples are much less forgiving with significant alterations in serum nucleosome concentrations noted after samples sat at room temperature for 45 minutes or longer, it is highly recommended that plasma is used to measure nucleosome concentrations.

There was no statistical difference between the EDTA and citrate plasma samples. Given that the samples used for this portion of the study are all from healthy dogs, it is expected that they will have relatively low concentrations of nucleosomes. Nucleosomes in healthy dogs are primarily released by white blood cells as they become activated or as they die. Plasma contains factors that serum does not such as clotting factors and vitamins, which may contribute to the stability of the white blood cells in the plasma samples when compared to serum [23, 24]. Additionally, both EDTA and sodium citrate bind calcium as a means of preventing coagulation. However, calcium also plays an integral role in the activation of white blood cells [25]. The lack of free calcium in plasma tubes may contribute to lower concentrations of nucleosomes in these sample types. Additionally, both EDTA and sodium citrate inhibit DNAse activity at fairly low concentrations [1, 26], which may also add to the stability of nucleosome concentrations within these samples.

Interestingly, the serum separator tubes had significantly lower nucleosome concentrations than serum from the no additive red top tubes. This may be related to the physical separation of the serum from the white blood cells during and after centrifugation. Even more interesting is the fact that the mean nucleosome concentration from the sodium citrate tubes was about 1/3 of the mean nucleosome concentration in the EDTA tubes. One explanation of this may be that EDTA is a much more efficient calcium chelator than citrate requiring 1/5 of the molar concentration to prevent gross coagulation [27]. Because of this, microplatelet clumps form within citrate plasma due to low level platelet activation in sodium citrate tubes [27]. It is possible that nucleosomes are binding or becoming entrapped in the microplatelet clumps within these tubes, lowering the number of free nucleosomes available for antibody binding within the samples.

Though there was slightly less variability in the citrate plasma samples, the higher concentrations of nucleosomes in EDTA makes this sample type more desirable. Either plasma sample was determined to be superior to serum and an effective way to repeatedly measure nucleosome concentrations in dogs. Given the high prevalence of EDTA tubes in general veterinary practice compared to sodium citrate tubes and the presumed enhanced capture of nucleosomes in EDTA plasma, the determination was made to use plasma (EDTA plasma with or without citrate plasma) for the majority of the assays moving forward.

Taken together, based on these results, it was determined that collecting samples in EDTA tubes and plasma isolation is the optimal method for evaluating nucleosomes in canine patients, furthermore, this is consistent with sample collection recommendations for humans.

The ability to ship samples enables collection to occur at individual veterinary offices, from which they can be sent to a centralized location for testing and analysis. Thus, we evaluated whether shipping the samples on ice changed the nucleosome concentrations in either EDTA or citrate plasma. In both cases those samples shipped on ice had less variability and lower nucleosome concentrations than those stored at room temperature. These lower

concentrations were more in line with time 0 nucleosome concentrations in the first two assays. Given that these samples were collected from the same dogs (paired samples from the same tube were stored either on ice or at room temperature) and that they had been centrifuged in the same tube before these paired aliquots were prepared, it is not possible for the room temperature samples to truly have higher nucleosome concentrations than samples stored on ice, rather it more likely that there is a minor temperature associated conformational change that enables better access of the antibody to the nucleosome. These differences were not statistically significant and this finding is not specific to our study as increased levels of DNA have been seen in biobank samples after long term storage as a result of protein disassociation allowing more DNA to be available for PCR amplification [28]. Thus, to ensure the most accurate results are generated it is recommended that samples be shipped on ice as these concentrations were most similar to time 0 nucleosome concentrations in other assays.

Nucleosomes are fairly stable in plasma after centrifugation. In these plasma samples, nucleosome concentrations were fairly stable after 3 freeze thaw cycles, with a noticeable decrease in nucleosome concentrations occurring in 2 of 6 samples by the 4th freeze/thaw cycle. The process of freezing and thawing has been shown to degrade protein and DNA and has even been published as a method of buffer free protein isolation from exosomes and other cell free DNA components [29]. However, samples may be safely be used and refrozen 2–3 times before the quality of the sample is compromised.

Interestingly, samples collected from fasting dogs had more consistent concentrations of nucleosomes for nearly all of the dogs and a greater variation in nucleosome concentration was seen after dogs were fed. Studies have shown that folic acid supplementation can affect the DNA methylation profile in mice, however, to date, no studies have been performed to determine if diet can alter the nucleosome content in mammals [30]. A high body mass index has been associated with elevated concentrations of circulating nucleosomes in humans [22, 31], however, no studies examining the effect of fasting on circulating nucleosome concentrations could be found in any species. Given this variability, it is recommended that any future samples drawn for dogs be fasting samples to limit the amount of variation seen.

Several of the groups analyzed in this manuscript were quite small which may have under or over-estimated differences between the groups. Additional animals should be compared to further validate some of the changes seen in the different processing and handling variables.

Finally, nucleosome concentrations were evaluated in client owned dogs presenting with naïve lymphoma and compared to the healthy dogs used in earlier assays within this study. All dogs diagnosed with lymphoma were fasted as part of our standard clinical recommendation for new patients. The samples collected from healthy dogs and assayed over a variety of time points were not all fasted samples. There was no significant difference detected in nucleosome concentrations for the dogs with lymphoma across any of the time points, however, there was a significant difference between the mean nucleosome concentrations from the dogs with lymphoma when compared to the healthy controls. Of the 13 dogs with lymphoma, only one had a mean nucleosome concentration that was similar to that seen in the control group (mean of this one dog was 23 ng/mL). The other 12 dogs had means that were much higher than what was found in the healthy control population. Elevated concentrations of cfDNA have been reported in dogs with cancer, however, this is the first time, to the authors' knowledge, that elevated nucleosome concentrations have specifically been reported in dogs with cancer [13, 32, 33]. Though this initial finding is promising, the small number of cases and use of only one type of cancer in this population, warrants further investigation before determining the utility of plasma nucleosome concentrations as a diagnostic or prognostic tool in veterinary oncology.

## Conclusions

Very little is known about nucleosomes in the cfDNA compartment in healthy or ill canines. The data presented here provides a better understanding of what this compartment typically looks like in healthy dogs and how simple variables, such as feeding or processing time can significantly alter the plasma nucleosome concentration in dogs. Regarding sample optimization for further analysis in healthy or ill dogs, the authors recommend using plasma rather than serum from fasted patients whenever possible. It is also important to process those samples within 60 minutes of collection (ideally 30 min whenever possible). If shipping these samples, it is recommended that samples ship over ice for the most consistent nucleosome concentrations. Regarding nucleosome concentrations in cancer patients, this preliminary work suggests that nucleosome concentrations may be elevated in some patients with cancer. Additional work is needed to determine the utility of measuring circulating nucleosome concentrations as a diagnostic or prognostic tool.

## Acknowledgments

The authors would like to acknowledge Spectrus Corporation, especially Dr. Michael Ziebell, for their work validating the H3.1 detection antibody for use in canines as well as the Fred and Vola Palmer Chair in Comparative Oncology for funding support of this work.

## Author Contributions

**Conceptualization:** Heather Wilson-Robles.

**Data curation:** Heather Wilson-Robles, Tasha Miller, Jill Jarvis.

**Formal analysis:** Heather Wilson-Robles, Terry Kelly, Thomas Bygott, Nathalie Hardat.

**Funding acquisition:** Heather Wilson-Robles, Jason Terrell, Nathan Dewsbury, Gaetan Michel.

**Investigation:** Heather Wilson-Robles, Jill Jarvis.

**Methodology:** Heather Wilson-Robles, Tasha Miller.

**Project administration:** Heather Wilson-Robles, Jill Jarvis.

**Resources:** Heather Wilson-Robles, Tasha Miller, Jill Jarvis, Jason Terrell, Nathan Dewsbury, Terry Kelly, Marielle Herzog, Thomas Bygott, Gaetan Michel.

**Software:** Heather Wilson-Robles, Thomas Bygott, Nathalie Hardat.

**Supervision:** Heather Wilson-Robles.

**Validation:** Heather Wilson-Robles, Tasha Miller.

**Visualization:** Heather Wilson-Robles.

**Writing – original draft:** Heather Wilson-Robles, Nathan Dewsbury, Terry Kelly.

**Writing – review & editing:** Heather Wilson-Robles, Jason Terrell, Nathan Dewsbury, Terry Kelly, Marielle Herzog, Thomas Bygott, Nathalie Hardat, Gaetan Michel.

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
