## [Decision Letter · Decision Letter 0]

22 Jul 2020

PONE-D-20-20189

Evaluation of nucleosome concentrations in healthy dogs and dogs with cancer

PLOS ONE

Dear Dr. Heather Wilson-Robles,

Thank you for submitting your manuscript to PLOS ONE. After careful consideration, we feel that it has merit but does not fully meet PLOS ONE’s publication criteria as it currently stands. Therefore, we invite you to submit a revised version of the manuscript that addresses the points raised during the review process.

We look forward to receiving your revised manuscript.

Kind regards,

Yi Cao

Academic Editor

PLOS ONE

Journal Requirements:

"The authors would like to acknowledge Spectrus Corporation, especially Dr. Michael

Ziebell, for their work validating the H3.1 detection antibody for use in canines as

well as the Fred and Vola Palmer Chair in Comparative Oncology for funding

support of this work."

"- HWR

- N/A

- This work was supported by an endowed chair held by HWR.

- N/A"

Additionally, because some of your funding information pertains to commercial funding, we ask you to provide an updated Competing Interests statement, declaring all sources of commercial funding.

In your Competing Interests statement, please confirm that your commercial funding does not alter your adherence to PLOS ONE Editorial policies and criteria by including the following statement: "This does not alter our adherence to PLOS ONE policies on sharing data and materials.” as detailed online in our guide for authors  http://journals.plos.org/plosone/s/competing-interests.  If this statement is not true and your adherence to PLOS policies on sharing data and materials is altered, please explain how.

Please include the updated Competing Interests Statement and Funding Statement in your cover letter. We will change the online submission form on your behalf.

3. Thank you for stating the following in the Competing Interests/Financial Disclosure* (delete as necessary) section:

"I have read the Journal's policy and the authors of this manuscript have the following competing interests: HWR is a paid consultant of Volition Veterinary. Volition Veterinary is a joint venture between Belgian Volition and Texas A&M University. Belgian Volition & Volition America are commercial developers of Nu.Q™ assays. The remaining authors have no conflicts of interest to declare."

We note that one or more of the authors are employed by commercial companies: Volition America & Volition Veterinary Diagnostic Development, Belgian Volition SPRL.

3.1. Please provide an amended Funding Statement declaring this commercial affiliation, as well as a statement regarding the Role of Funders in your study. If the funding organization did not play a role in the study design, data collection and analysis, decision to publish, or preparation of the manuscript and only provided financial support in the form of authors' salaries and/or research materials, please review your statements relating to the author contributions, and ensure you have specifically and accurately indicated the role(s) that these authors had in your study. You can update author roles in the Author Contributions section of the online submission form.

3.2. Please also provide an updated Competing Interests Statement declaring this commercial affiliation along with any other relevant declarations relating to employment, consultancy, patents, products in development, or marketed products, etc. 

Reviewers' comments:

Reviewer's Responses to Questions

**Comments to the Author**

1. Is the manuscript technically sound, and do the data support the conclusions?

Reviewer #1: Yes

Reviewer #2: Yes

2. Has the statistical analysis been performed appropriately and rigorously? 

Reviewer #1: Yes

Reviewer #2: Yes

3. Have the authors made all data underlying the findings in their manuscript fully available?

Reviewer #1: Yes

Reviewer #2: Yes

4. Is the manuscript presented in an intelligible fashion and written in standard English?

Reviewer #1: Yes

Reviewer #2: Yes

5. Review Comments to the Author

Reviewer #1: In this work,author use the Nu.Q TM H3.1 assay and determine nucleosomes in the blood of healthy canines as well as those with cancer,This is a pithy and innovative technique. But, it's a little problem for most ordinary people how to use sample collection and processing method appropriately，If possible, the author can optimize the sampling method, so that more people can use the technology correctly.

Reviewer #2: This manuscript by Wilson-Robles H, et al, reported the nucleosome data in the blood of dogs. This study tested and optimized the ELISA based technology to measure the blood nucleosomes and found for the first time that blood nucleosomes were increased in the dogs with lymphoma. Overall this study provided interesting data and valuable conditioning for accurate measurements of nucleosomes and sampling concerns. It is of interest for the development of this method for human use.

Concerns:

This figures needs to be reorganized for publication and the figure numbers were not matched with that in text.

May modify the title to lymphoma. "Cancer" is too general for this study that investigated lymphoma only.

6. PLOS authors have the option to publish the peer review history of their article (what does this mean?). If published, this will include your full peer review and any attached files.

Reviewer #1: No

Reviewer #2: **Yes: **Deliang Cao

---

## [Author Response · Author response to Decision Letter 0]

3 Aug 2020

The authors are very appreciative of the comments provided and have worked to address each comment made. 

Journal Requirements:

1. All figure names have been changed to fit the format requirements. For example, Fig 1A 300dpi.tiff has been changed to Fig 1A.tiff.

2. We appreciate the clarification regarding funding. We has incorrectly interpreted this section as a place to list extramural grants. This has been corrected and the acknowledgment section has been altered so that only the acknowledgement for Spectrus Corporation. The Chair position that Dr. Wilson-Robles holds has been added to the Funding Statement. 

Additionally, all corporate funding information has been provided in the Funding Statement and the following statement has been added to the Competing Interests Statement, "This does not alter our adherence to PLOS ONE policies on sharing data and materials." I can assure you that this statement is true and does apply to all data presented in this manuscript. 

3.1 Thank you for this clarification regarding the description of commercial developers as authors. The funders of the study did not play a role in study design or data collection (that work was performed by Wilson-Robles, H, Miller, T and Jarvis, J). However, the correlation statistics were performed by Bygott, T and Hardat, N (Belgian Volition). Additional authors assisted with manuscript preparation and organization (Michel, G, Herzog, M- Belgian Volition; Terrell, J, Kelley, T- Volition America; Dewsbury, N- Volition Veterinary). The author roles have been updated in the Authors contribution section.

We have also provided the following statement in the Funding Statement section: The funder provided support in the form of salaries for the authors (HWR, TM, JJ) as well as statistical analysis for correlation statistics (TB, NH) and preparation of the manuscript (TK, JT, ND, GM, MH). The funder did not have any additional role in the study design, data collection or the decision to publish.

3.2 The following statement has been added to the Competing Interests Statements, "This does not alter our adherence to PLOS ONE policies on sharing data and materials." 

Additionally, the cover letter has been updated with the revised Funding Statement and Competing Interests statements. 

4. These statements have been removed and the manuscript updated. 

5. Reviewer one comments: a summary statement has been clarified in the conclusion section describing the optimal sampling method for ease of use. 

Concerns:

The figures have been adjusted so that they are clear in the text and associated with the correct figure. 

The title has been altered to say lymphoma rather than cancer as suggested.

---

## [Decision Letter · Decision Letter 1]

13 Aug 2020

Evaluation of nucleosome concentrations in healthy dogs and dogs with cancer

PONE-D-20-20189R1

Dear Dr. Heather Wilson-Robles,

We’re pleased to inform you that your manuscript has been judged scientifically suitable for publication and will be formally accepted for publication once it meets all outstanding technical requirements.

Kind regards,

Yi Cao

Academic Editor

PLOS ONE

Additional Editor Comments (optional):

Reviewers' comments:

Reviewer's Responses to Questions

**Comments to the Author**

1. If the authors have adequately addressed your comments raised in a previous round of review and you feel that this manuscript is now acceptable for publication, you may indicate that here to bypass the “Comments to the Author” section, enter your conflict of interest statement in the “Confidential to Editor” section, and submit your "Accept" recommendation.

Reviewer #1: All comments have been addressed

Reviewer #2: (No Response)

2. Is the manuscript technically sound, and do the data support the conclusions?

Reviewer #1: Yes

Reviewer #2: (No Response)

3. Has the statistical analysis been performed appropriately and rigorously? 

Reviewer #1: Yes

Reviewer #2: (No Response)

4. Have the authors made all data underlying the findings in their manuscript fully available?

Reviewer #1: Yes

Reviewer #2: (No Response)

5. Is the manuscript presented in an intelligible fashion and written in standard English?

Reviewer #1: Yes

Reviewer #2: (No Response)

6. Review Comments to the Author

Reviewer #1: In this work, The Nu.Q TM technology measures circulating nucleosome levels，The Nu.Q TM is used to determine if it can accurately detect nucleosomes in the blood of healthy canines as well as those with cancer. This method is innovative and practical.

Reviewer #2: (No Response)

7. PLOS authors have the option to publish the peer review history of their article (what does this mean?). If published, this will include your full peer review and any attached files.

Reviewer #1: No

Reviewer #2: **Yes: **Deliang Cao

---

## [Editor Report · Acceptance letter]

19 Aug 2020

PONE-D-20-20189R1 

Evaluation of nucleosome concentrations in healthy dogs and dogs with cancer 

Dear Dr. Wilson-Robles:

I'm pleased to inform you that your manuscript has been deemed suitable for publication in PLOS ONE. Congratulations! Your manuscript is now with our production department. 

Kind regards, 

on behalf of

Dr. Yi Cao 

Academic Editor

PLOS ONE